# Clinical Pharmacist-Led Collaboration of Multiple Clinical Professions Model Focusing on Continuity of Pharmacotherapy: Japanese Version of the Lund Integrated Medicines Management (LIMM) Model

**DOI:** 10.3390/pharmacy12060184

**Published:** 2024-12-05

**Authors:** Rie Sato, Syuichi Aoshima, Tommy Eriksson

**Affiliations:** 1Center for Primary Health Care Research, Department of Clinical Sciences, Faculty of Medicine, Lund University, CRC 28-11, Jan Waldenströms Gata 35, Box 50332, SE202-13 Malmö, Sweden; 2Department of Emergency and Critical Care Medicine, Faculty of Medicine, Shimane University, Izumo City 693-8501, Shimane, Japan; 3Medical Corporation Tokujin-kai, Nakano Hospital, 8-1, Iwai-cho, Tochigi City 328-0052, Tochigi, Japan; syuichiao@gmail.com; 4Association for Appropriate Healthcare Decision-Making and Practice, Higashihiroshima City 739-0024, Hiroshima, Japan; 5Department of Biomedical Science and Biofilm, Research Center for Biointerfaces, Faculty of Health and Society, Malmö University, SE205-06 Malmö, Sweden; tommy.eriksson@mau.se

**Keywords:** pharmacotherapy, continuity of care, primary health care, clinical pharmacists, the Lund Integrated Medicines Management (LIMM) model, Sweden

## Abstract

(1) Background: In general, it is known that continuity of care can contribute to an increase in patient satisfaction, reduce health care costs, and improve patient outcomes. A guarantee of continuity in pharmacotherapy is a big challenge facing Japanese health care as a system that encourages cooperation/collaboration for pharmacists with other health care professions is currently lacking. (2) Method: This is a narrative review. (3) Results: The Lund Integrated Medicine Management (LIMM) model describes a systematic approach to individuals and was developed in Sweden to optimize pharmacotherapy among elderly inpatients. The aim of the LIMM model is to provide patients with continuous pharmacotherapy at different levels of care. The LIMM model, in which a clinical pharmacist is the catalyst and leads other health care professions in completing the process, has the potential to reduce potentially inappropriate prescriptions, reduce rehospitalization risk, unscheduled hospital revisits due to problems related to medications, reduce total medical expenditure, and provide a comprehensive understanding of patients’ conditions of taking medicine. (4) Conclusions: Introducing a framework such as Sweden’s LIMM model, anchored by clinical pharmacists, could provide a good opportunity to promote collaborations among different health care professionals and improve continuity in pharmacotherapy.

## 1. Introduction

In Japan, it is difficult for many medical health care professionals, including pharmacists, to actively cooperate/collaborate with different specialists, e.g., physicians and hospital pharmacists. This is because community pharmacies are independent of hospitals in which physicians and hospital pharmacists work, and it is difficult to share patients’ care information, including pharmacotherapy. Adding to this, it is difficult for hospital pharmacists to have close communication with physicians, as their work is independent of each other. Furthermore, they tend to avoid overlapping their specialties and work and to respect others “too much”. As just described, a guarantee of continuity in pharmacotherapy is one of the biggest challenges facing Japanese health care as a system that encourages cooperation/collaboration for pharmacists with other health care professions, including physicians, is currently lacking.

### 1.1. History of Pharmacist Identity in Japan

Japanese medicine has improved having been previously modeled on oriental medicine, in which the border between medicine and pharmacotherapy was generally vague and did not have a clear distinction. Before the Meiji era (end of the 1800s), physicians oversaw pharmaceutic management as a matter of course [1]; therefore, there was no pharmacist as a specialist in dispensing medicines. After the Meiji era, the separation of medical practice and pharmaceutical compounding [2] started to attract attention officially for the first time, and the qualifications and social status of pharmacists were defined publicly in 1889. However, it took a while to officially realize the separation of medical practice and pharmaceutical compounding. After 1997, it has been ingrained among people that outpatients obtain their medications according to a prescription from pharmacies independent of clinics or hospitals, and that pharmacists are in charge of preparing medicine.

In primary health care services, the family pharmacist system, which is called “kakaritsuke-yakuzaishi”, was started in 2016 to encourage patients to have their personal pharmacists and pharmacies to support continuous care [3,4]. However, there are discrepancies between patients’ needs and pharmacists’ intentions of “what pharmacists want to do for patients” in a clinical context. It could be said that the formation of pharmacist identities in clinical practice is being developed, which is also being seen worldwide [5,6,7].

### 1.2. Lack of Continuity of Pharmacotherapy in Japan

In Japan, the concept of primary care is not well established, and many patients tend to directly consult specialists. As a result, patients with multiple medical conditions receive separate prescriptions from different specialized medical institutions [8] and have their medications dispensed at different pharmacies. This situation makes it extremely difficult to centrally track and effectively manage patients’ medication information.

In 2016, the Japanese Ministry of Health, Labor and Welfare published “Vision of Pharmacies for Patients” [9], which indicated three functions that family pharmacists and pharmacies should obtain: central and continuous comprehension of medication information, 24 h patient support and home medical care support, and cooperation with medical facilities such as hospitals and clinics.

Central and continuous comprehension of medication information to provide continuous care to patients is one of the biggest challenges as it could be strongly associated with patients’ prognosis and adverse events [10,11,12]. For example, it is difficult for pharmacists and pharmacies to follow a patient’s health status when the patient is hospitalized and for outpatient follow-up, or if the patient changes their pharmacies or clinics/hospitals by their own choices. It is also identified that “poly-doctoring”, where patients see multiple specialists; as a result, patients going to multiple different pharmacies to receive their medication has emerged as one of the most crucial aspects of fragmentation of care and has a significant relationship with polypharmacy and increased medical expenditure at outpatient clinics [13].

Generally, “continuity of care” indicates a process in which a care team, which consists of medical specialists and a physician as a leader, is getting involved with patient care continuously, pursuing a common objective to realize qualified and cost-effective practice [14]. Practice anchored with continuity of care could be realized to improve patient satisfaction, reduce medical expenses, and improve patients’ prognoses [15,16,17,18]. To realize these continuities, broad knowledge, which has no boundary between different health care professions and organizations, and the sharing of correct information are essential [19].

Japanese pharmacists can be classified as hospital pharmacists and pharmacy pharmacists. To provide continuous medical care to the patients in the community, both pharmacy pharmacists and hospital pharmacists should play major roles by cooperating/collaborating with other health care professionals. However, the opportunities are limited even where pharmacy pharmacists and hospital pharmacists can collaborate with each other and continuity of care has not been fully achieved [20,21,22]. One of the reasons for this situation is that hospital pharmacists and community pharmacists belong to different professional organizations and conduct their academic activities separately. While most hospital pharmacists in Japan are members of the Japanese Society of Hospital Pharmacists (JSHP) [23], most community pharmacists belong to the Japan Pharmaceutical Association (JPA) [24]. Moreover, in Japan, where primary care functions are inadequate, it is challenging for hospital pharmacists to collaborate with multiple community pharmacists, and there are no established methodologies or systems for such collaboration.

To our knowledge, there has not been research conducted that focused on the continuity of care, especially in a pharmaceutical context. In this review, we would like to introduce the Lund Integrated Medicines Management (LIMM) model, which is a framework that is anchored by clinical pharmacists emphasizing continuity of care, and discuss the influence of their work according to the LIMM model on patient outcomes and the possibility to introduce the Swedish LIMM model to Japan.

## 2. Introduction of Lund Integrated Medicines Management (LIMM) Model

For the optimization of drug treatment for an individual, a systematic approach is essential [25]. An integrated medicine management (IMM) service has been developed in Northern Ireland, which involves pharmaceutical care at admission, during the hospital stay and at discharge [25]. The Lund Integrated Medicine Management (LIMM) model, which describes a systematic approach to individuals, was developed in Sweden to optimize pharmacotherapy among elderly inpatients [26]. This was expected to serve as a catalyst for cooperation/collaboration among health care professionals that is anchored by clinical pharmacists. A clinical pharmacist is a health care professional who works directly with physicians, other health professionals, and patients to ensure that medications prescribed for patients contribute to the best possible health outcomes. They are involved in direct patient care, optimizing medication use, and promoting health, wellness, and disease prevention [27]. The aim of the LIMM model is to provide patients with continuous pharmacotherapy in different levels of care, e.g., hospital and primary care.

The model covers several aspects of medication care, from appropriate prescriptions, how drugs are taken or not taken by patients, drug-related problems, to communication between care levels and with the patient [26]. The LIMM model covers these aspects comprehensively, which is shown in Figure 1 [28]. The LIMM model is team-based and consists of three parts: systematic medication reconciliation, medication review, and oral and written communication, pursuing continuous care throughout primary and hospital care. In the LIMM model, there are three activities throughout hospitalization.

At admission, clinical pharmacists conduct the admission medication reconciliation with the patient, using the LIMM Medication Interview Questionnaire. It consists of three parts: Part 1 is an identification of the most accurate patient medication list. Part 2 focuses on the patient’s practical handling and knowledge of the medications and adherence to the medical regimen. Part 3 is an additional detailed interview concerning the patient’s adherence to the medical regimen and personal attitude regarding medications. Parts one and two are mandatory and Part three is performed if the patient is capable and willing to participate.

During their hospital stay, structured medication reviews and monitoring by clinical pharmacists to identify drug-related problems, and symptom assessment by nurses or clinical pharmacists are conducted at regular intervals for each patient. Physicians also lead the multi-professional team, organize a treatment plan based on the symptom assessments, medication reviews, and reconciliation results, and discuss the drug-related problems that pharmacists consider to be the most relevant. Patients are followed up at least twice a week to identify other new drug-related problems and to monitor previously identified issues. All data are documented on the LIMM Medication Review Form. Pharmacotherapy is a dynamic process where the course of the disease, the effects of medications, etc. must be continuously evaluated. Depending on what needs to be monitored, this is carried out by the most suitable professional.

Upon discharge, patients and primary or community care personnel receive the discharge medication reconciliation information from the physician, using the LIMM Discharge Information Form, including a Medication Report and a Medication List, which is a list of current drugs, doses, and indications. Sometimes clinical pharmacists document quality control of discharge medication reconciliation, using the LIMM Quality Control form for Discharge Medication Reconciliation. In the LIMM model, discharge medication reconciliation is a process where structured written information is provided to the patient and sent to the responsible primary care physician and, if applicable, to community care nurses. However, the community pharmacy organization has stated that they do not want to handle this information due to the significant workload involved in comparing it with existing prescriptions. It is still a challenge to avoid the discrepancies between the discharge medication list in their discharge summaries and the medications actually taken by the patients after discharge [29]. It is essential for physicians to conduct the discussion about the medications well in advance of the discharge and for clinical pharmacists to follow up with the patients after their discharge [29]. The physician is responsible for prescribing medications, while the pharmacist has the expertise and competence to support the physician and the patient in this process. When the LIMM model was planned, it was considered important that the physician continues to take this responsibility. Since the discrepancies between discharge medication and medications actually taken by the patients after discharge seem problematic, a very recent feasibility study, before a randomized-controlled trial, proposed a solution [29]. The pharmacist performs a discharge medication interview with the patient and addresses discrepancies with the physician who directly corrects medication list errors.

A clinical pharmacist is the catalyst and leads other health care professionals in completing the process [28], although other professionals in the team are fundamental for success. The LIMM model process is team-based, where communication and sharing of information is essential. The process should be followed up on its quality, researched, and developed stepwise.

Table 1 summarizes the significant studies and their findings regarding the expected clinical outcomes of interventions of the LIMM model. The LIMM model showed a positive effect on the appropriateness of pharmaceutic therapy according to the Medication Appropriateness Index (MAI), which is a validated instrument to measure the pharmacological and economic aspects of prescribing appropriateness [26]. The benefit of the LIMM model in the process and some outcomes, in which high-quality methods and design for a team approach were used, has been established [28]. A systematic analysis at Lund University Hospital found that the LIMM model reduced health care contacts and hospital readmissions due to medication errors by 50%, and pharmacists’ work could save physicians’ and nurses’ working hours [30]. It has been shown that the LIMM model can be implemented in similar settings and the outcomes are guaranteed [28]. The LIMM model has had a large impact on pharmacists’ education as an educational platform for training in different contexts, e.g., pharmacotherapy, communication, and clinical skills, which could improve the core competencies of pharmacists [31].

## 3. Discussion

### 3.1. Strong Points of the LIMM Model

As mentioned previously, the LIMM model is operated by the cooperation/ collaboration of different health care professions anchored by clinical pharmacists. The LIMM model has been reported to have possibilities to reduce potentially inappropriate prescriptions [25,26], reduce rehospitalization risk [38], unscheduled hospital revisits due to problems related to medications [26], the total medical expenditure [40], and grasping comprehensively patients’ conditions of taking medicine [36]. A health economic study showed that investing €39 in clinical pharmacist time could save €340 in medical care at hospitals and in primary care [40].

Generally, the effect size of the intervention of deprescribing in potentially inappropriate medications is thought to be limited. There are few reports that indicate the improvement of clinical outcomes or patients’ prognoses due to the interventions of deprescribing, even though they could reduce the number of prescribed medications [41]. Especially in primary care settings, the interventions of deprescribing are said that it could be impossible to reduce the number of prescribed medications [42]. It could be argued that the implementation of interventions only focused on potentially inappropriate medications would not improve patients’ prognoses.

On the other hand, interventions that provide not only medication reviews but also involve cooperation between different health care professions, motivational interviews, and individual care plans for patients would reduce the number of rehospitalizations as well as mortality risks [43,44]. Therefore, the improvement that the LIMM model has achieved is due to the cooperation/collaboration of different specialists anchored by clinical pharmacists and continuity of care, due to not only appropriate reviews of prescribed medications.

### 3.2. The Possibility of Implementation of LIMM Model in Japan, the Japan Integrated Medicines Management (JIMM)

In Japan, it is difficult for pharmacists to cooperate/collaborate with physicians closely due to the difficulties in sharing the same patient information. The information that physicians convey and emphasize to their patients and what they deem to be important are often not the same. Pharmacists show excessive consideration to physicians in avoiding overlap of their specialties and work, and they have different priorities towards treatment processes [45,46].

Pharmacotherapy is one of the cores of medical service. It is reasonable to say that the continuity of pharmacotherapy is one of the most important factors in providing patients with continuity of care. However, it is not common practice to share medical records with different medical facilities in Japan so continuity of pharmacotherapy is one of the biggest challenges, especially for patients who see different doctors at different medical facilities [13]. To solve these problems, it is thought to be needed to focus on the dynamic evaluation of prescription, not to focus on fragmented evaluations at a certain point in time. As it is important to focus on the dynamic prescription trajectory of the health condition changing over time, the framework of the cooperation/collaboration of different health care professions anchored by pharmacists, focusing on the continuity of pharmacotherapy, is thought to be essential.

It is also important that pharmacists should play the main role in operating the framework. In fact, clinical pharmacists are evaluated highly by other specialists and patients in LIMM, and about 90% of recommendations that clinical pharmacists raised have been accepted positively by physicians [32,33]. The introduction of a framework like LIMM to Japan has the possibility to bring improvements in high-quality cooperations/collaborations, more optimized implementation of pharmaceutical care available to patients individually, and consequent improvement of Japanese public health. For this, the pharmacists have to be formally trained in university courses focusing on evidence-based medicine, patient-centered care, communication skills, and national and regional pharmacotherapy guidelines to be able to effectively perform medication reconciliation and -review to identify, solve, and prevent drug-related problems.

The LIMM model is expected to serve as a catalyst for cooperation/collaboration among health care professionals, particularly centered around pharmacists. Rather than focusing primarily on screening inappropriate prescriptions, this approach aims to resolve drug-related problems. In fact, medication adherence is thought to be related to not only patient awareness but also health system technology and better collaborations between health care professionals [47]. Collaborations between primary care pharmacists and other fields, such as oncology, have improved medication management [48]. It is also shown that the appropriate interprofessional collaboration to support medication adherence for optimizing treatment outcomes among oncology patients who use Trametinib is needed [49]. In this sense, the LIMM model represents a different dimension from just mechanical drug inappropriateness evaluation criteria.

In Japan, the aging society with the advancement of medical care and increasing health care demands have led to a growing workload for physicians, which has become a social issue. In this context, task-shifting, which involves transferring certain responsibilities to non-physician health care professionals to reduce long working hours, has gained attention [8]. The implementation of LIMM is expected to promote task-shifting, potentially reducing physicians’ workload while improving the quality of medical services.

The LIMM model has been implemented fully in Oman [50]. Activities to spread it in Europe have been taken by the European Association of Hospital Pharmacy [51]. Japan has its own health care system, culture, history of its own medicine, and regulations, and there are many organizations that have the potential power to influence society. It is thought to be very important to refer to the other countries’ successful implementation, but at the same time, it is also very important to consider Japanese social contexts before implementing this approach into Japanese society.

The main purpose of this review is to introduce the LIMM model to Japan and encourage readers to know about it. The implementation would be the next step after this concept is accepted by Japanese society. However, it is a very new concept in Japan and the authors believe that this review could influence Japanese pharmacotherapy.

## 4. Conclusions

Cooperation and collaboration among multiple health care professions and continuity of care are important factors that can improve clinical outcomes. However, the lack of continuity of pharmacotherapy due to difficulties of collaborations between physicians and pharmacists is one of the most important challenges for Japanese public health. An introduction of a framework, such as Sweden’s LIMM model, could provide a good opportunity to promote collaborations among different health care professionals, which is anchored by pharmacists. Continuity improvements in pharmacotherapy could then be expected.

## Figures and Tables

**Figure 1 pharmacy-12-00184-f001:**
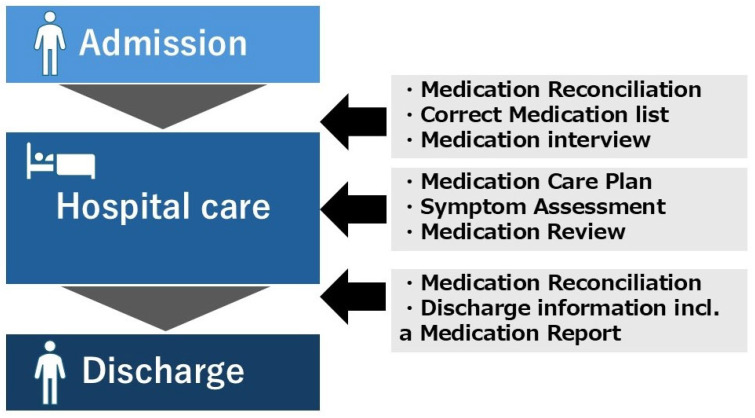
The Lund Integrated Medicines Management (LIMM) model. LIMM model is team-based and consists of three parts: systematic medication reconciliation, medication review, and oral and written communication, pursuing continuous care throughout primary and hospital care. The copyright has been approved by the author and BMJ journal.

**Table 1 pharmacy-12-00184-t001:** Results from the studies concerning the LIMM model.

Questions	Results	Reference
Are patient drug-related problems (DRPs) identified systematically?	The clinical pharmacist identifies 6–10 DRPs per patient.	Bergkvist 2011 [32], Bondesson 2012 [33]
Are many DRPs missed?	The number of unidentified DRPs decreased from 9 to 1 by a pharmacist intervention.	Bondesson 2013 [34]
Are recommendations by the clinical pharmacist used?	In total, 90% of recommendations were accepted and implemented by the physician.	Bondesson 2012 [33]
Are the recommendations by clinical pharmacists important?	In total, 49% of recommendations were ranked as ‘significant’ or higher, and 83% as ‘somewhat significant’ or higher.	Bondesson 2012 [33]
What does the medication interview add?	A medication interview identifies errors in the medication list, and patient problems with knowledge, handling, compliance, and attitudes.	Bondesson 2009 [35], Hellström 2012 [36]
What does the Medication Report add?	Medication errors and care contacts due to the errors are reduced by 50%.	Midlöv 2008 [37]
Can error rates at discharge be further reduced?	Quality control and direct feedback to a physician of patient discharge information by a pharmacist decreased error rates at discharge and also 2 weeks after discharge by more than 45%.	Bergkvist 2009 [25] Al-Musawi 2024 [29]
Does the treatment improve?	Medication appropriateness Index (MAI) improves.	Bergkvist 2009 [25], Bondesson 2012 [33]
What do the physicians and nurses think about the LIMM and clinical pharmacists’ contribution?	The benefits for the patient and the health care personnel were estimated to be very high, with a median of 5–6 on a 6-number Likert scale.	Bergkvist 2012 [32], Bondesson 2012 [33]
Is there a patient benefit?	Care contacts due to DRP errors are reduced by 50% with the Medication Report.	Midlöv 2008 [37]
	Hospital readmission due to DRP further decreased with pharmacist interventions (54%, ARR = 6.4%, NNT = 16).	Hellström 2011 [26]
	Total emergency department visits were not affected in routine clinical practice.	Hellström 2012 [38]
Does the LIMM save time?	When a clinical pharmacist spent 1 h on patient activities, the physician and nurse saved 1.5–2 h at the hospital and 0.5–1 h in primary care	Eriksson 2012 [39]
What are the health-economic benefits?	LIMM generated both costsavings and higher utility to the patients. Investing €39 in clinical pharmacist time could save €340 in medical care at hospitals and in primary care.	Ghatnekar 2013 [40]

The LIMM model showed a positive effect on drug-related problems (DRPs), medication errors, Medication Appropriateness Index (MAI), care contacts due to DRP errors, hospital readmissions due to DRP, and total emergency department visits, working hours of physicians and nurses, and the cost of medical care at hospital and primary care.

## Data Availability

No new data were created or analyzed in this study. Data sharing is not applicable to this article.

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
