# Peer review of "Clinical Pharmacist-Led Collaboration of Multiple Clinical Professions Model Focusing on Continuity of Pharmacotherapy: Japanese Version of the Lund Integrated Medicines Management (LIMM) Model"

_pharmacy, 2024, doi:10.3390/pharmacy12060184_

Round 1
Reviewer 1 Report
Comments and Suggestions for Authors
Thank you for allowing me to read your opinions regarding the challenges of inter professional collaboration in Japan and the potential of the LIMM model to improve those challenges.
I can see that you are advocating for the LIMM model, what I cannot see is how this is more than opinion. Not necessarily an inappropriate opinion, but there is no new data here.
Additionally, there are some interesting choices in classifying pharmacists. I challenge you to describe a "non-clinical pharmacist." But, the term "clinical pharmacist" appears starting on line 119 and multiple times after that. If the implication is that community pharmacists or pharmacy pharmacists are not clinical, it begs the question what do you want them to do with the medication reconciliation created at discharge?
There is another technical challenge appearing on lines 131 and 132. Patients are following up at least twice a week to identify.... Followed up by who? How long do these twice weekly reviews continue? What is the benefit at week 5 that didn't exist at week 4?
Lastly, in a manuscript that appears to support the inclusion of pharmacists in the pharmacotherapy process, it is interesting that "it is essential for PHYSICIANS to conduct the discussion about the medication..." Again, why? If the pharmacists are the drug experts, why would it be "essential" for a physician to talk about the drugs?
Please review your work to determine if there is a better way to present your vision and your recommendations.
Comments on the Quality of English LanguageThere are a number of grammar challenges with this manuscript.
There are challenges with verb-tense (line 50, line 77, etc)
There are several interesting wording choices
"drug dispensation" is used line 54, line 57. I believe you mean dispensing, but I cannot be certain. You may actually be referring to the dispensation, or location of the drugs, but that doesn't fit the rest of the narrative well.
There is significant overuse of the word "however" with no benefit to the manuscript (line 56, line 62, line 93, line 139, etc). Consider removing this is all instances.
I am unable to understand what the phrase "continuous comprehension" means and I'm sorry, but I can offer to suggestions for correction. See line 69.
Reviewer 2 Report
Comments and Suggestions for Authors
This narrative review explores the challenge of achieving continuity of pharmacotherapy in Japan, where collaboration between pharmacists and other healthcare professionals is limited. The authors introduce the Lund Integrated Medicines Management (LIMM) model, a framework developed in Sweden that emphasizes clinical pharmacist-led collaboration across healthcare settings to optimize medication management. The review discusses the potential benefits of implementing a similar model in Japan, highlighting its potential to improve patient outcomes, reduce medication errors, and enhance pharmacist education.
General Concept Comments:
Relevance and Gap: The review addresses a critical gap in knowledge regarding continuity of pharmacotherapy in Japan and provides a timely and relevant discussion on the potential for collaborative models like LIMM to address this issue.
Comprehensiveness: The review covers the historical context of pharmacist identity in Japan, the current challenges in achieving continuity of pharmacotherapy, and the key features and benefits of the LIMM model. However, the review could benefit from a more detailed discussion of the specific barriers to implementing LIMM in Japan and potential solutions to overcome these barriers.
References: The references are generally recent and relevant, with a few exceptions. The review could benefit from including more recent studies on deprescribing interventions and their impact on patient outcomes.
In addition, the paper explores an innovative framework for enhancing pharmacotherapy through interdisciplinary collaboration. The following key references thus are recommended for this study:
Zhang, Jing et al. "Application of Once-Monthly Self-Reported ACT Questionnaire in Management of Adherence to Inhalers in Outpatients with Asthma." Patient preference and adherence vol. 14 1027-1036. 19 Jun. 2020, doi:10.2147/PPA.S176683.
Davidson, Arielle et al. "Managing medications for patients with cancer and chronic conditions..." Journal of Oncology Pharmacy Practice. [DOI: 10.1177/10781552241279303]
Hafez, Gaye et al. "Barriers and Unmet Educational Needs..." Journal of General Internal Medicine. [DOI: 10.1007/s11606-024-08851-2]
Ravix, Anne et al. "Population Pharmacokinetics of Trametinib..." Cancers. [DOI: 10.3390/cancers16122193]
These references are particularly relevant for several reasons. Firstly, it emphasizes medication adherence, which can be seen as a parallel expression of the continuity of pharmacotherapy that this paper aims to address. Secondly, it advocates for a collaborative model led by clinical pharmacists, highlighting its effectiveness in improving patient medication management. This supports the discussions within the paper regarding the cooperation and collaboration among various healthcare professionals, particularly the pivotal role of clinical pharmacists. Lastly, the cooperative strategies outlined in the article can provide practical insights for implementing the LIMM model effectively, further enriching the proposed framework. By integrating these perspectives, we can enhance our understanding of how structured collaboration can lead to better health outcomes in pharmacotherapy continuity.
Specific Comments:
Line 81: Consider clarifying the definition of “continuity of care” and the different types (information, management, relationship) to enhance clarity for readers unfamiliar with the concept.
Line 93: The statement “However, the opportunities are limited even where pharmacy pharmacists and hospital pharmacists can collaborate with each other and continuity of care has not been fully achieved” could be strengthened with specific examples or data supporting this claim.
Line 111: Consider expanding on the “several aspects of medication care” covered by the LIMM model to provide a more comprehensive overview.
Line 139: The challenge of discrepancies between discharge medication lists and actual patient medication use is significant. Consider discussing potential strategies to address this issue within the LIMM model.
Line 163: The review highlights the potential of the LIMM model to reduce potentially inappropriate prescriptions. Consider discussing the specific criteria used to identify inappropriate medications within the LIMM model and how this differs from other deprescribing interventions.
Line 188: The review mentions the importance of pharmacists playing a leading role in operating the framework. Consider discussing the specific training and skills required for clinical pharmacists to effectively lead such collaborations.
Line 223: Acknowledge the potential limitations of the study, such as the lack of empirical data on the implementation of LIMM in Japan and the potential for cultural differences between Sweden and Japan.
Additional Questions:
Have the authors considered the potential costs and resource implications of implementing a LIMM-like model in Japan?
How would the authors propose addressing the potential resistance from physicians or other healthcare professionals to the increased role of pharmacists in medication management?
What specific steps would be necessary to adapt the LIMM model to the Japanese healthcare system, taking into account cultural, regulatory, and organizational differences?
What is the potential for implementing the LIMM model in other countries?
Overall, this is a well-written and informative review that provides valuable insights into the challenges and potential solutions for achieving continuity of pharmacotherapy in Japan. With some minor revisions and further exploration of implementation considerations, this review could be a valuable contribution to the field.
Comments on the Quality of English LanguageMinor editing of English language required.
Round 2
Reviewer 1 Report
Comments and Suggestions for Authors
Thank you for the opportunity to review your revised manuscript. I appreciate the work you put forth to respond to the reviewers.
Page 5, Table 1 - - I understand this table and when I reformat it for review, I can read it easily. I would caution the authors that this will need careful review when you see the final, type-set version of your manuscript to assure that it is spaced correctly and makes sense.
Page 8 - - the added language here, describing what you aspire to have occur is very helpful in focusing your manuscript. Thank you
Comments on the Quality of English Language
No comments that have not already been provided